# Prospect of Bioactive Curcumin Nanoemulsion as Effective Agency to Improve Milk Based Soft Cheese by Using Ultrasound Encapsulation Approach

**DOI:** 10.3390/ijms24032663

**Published:** 2023-01-31

**Authors:** Uday Bagale, Ammar Kadi, Mostafa Abotaleb, Irina Potoroko, Shirish Hari Sonawane

**Affiliations:** 1Department of Food and Biotechnology, South Ural State University, Chelyabinsk 454080, Russia; 2Department of System Programming, South Ural State University, Chelyabinsk 454080, Russia; 3Department of Chemical Engineering, National Institute of Technology Warangal, Telangana 506004, India

**Keywords:** curcumin nanoemulsion, cheese, ultrasound, SEM, self-life and sensory analysis

## Abstract

The aim of this paper was to determine the effect of stabilized curcumin nanoemulsions (CUNE) as a food additive capable of directionally acting to inhibit molecules involved in dairy products’ quality and digestibility, especially cheese. The objects were cheeses made from the milk of higher grades with addition of a CUNE and a control sample. The cheeses were studied using a scanning electron microscope (SEM) in terms of organoleptic properties, such as appearance, taste, and aroma. The results show that the addition of CUNEs improved the organoleptic properties compared to the control cheese by 150% and improved its shelf life. The SEM study shows that formulation with CUNE promotes the uniform distribution of porosity. The CUNE-based cheese shows a better sensory evaluation compared to the emulsion without curcumin. CUNE-processed cheese provided better antioxidant and antimicrobial analysis than the control sample and offers added value to the dairy sector.

## 1. Introduction

Dairy products are an important sector of the food industry. Daily dairy consumption varies from 150 to 500 g per capita in different countries and is steadily increasing [1]. Processed cheese has gained more success, owing to a combination of economical available ingredients and better functional properties than other cheese [2,3]. However, processed cheeses have a problem with storage and shelf life. This problem can be minimized by the fortification of cheese with bio-active compounds. Bioactive compounds, such as essential oils, medical plants, and fruit extracts, have resulted in better versions of cheese [4,5,6,7]. The addition of bioactive compounds in processed cheese also affects the taste and the consistency [7].

A common antioxidant is curcumin, extracted from turmeric root [7,8,9,10,11]. Curcumin is a polyphenol, characterized by more than one phenolic group per molecule. Turmeric contains a wide variety of vitamins and other substances essential for the human organism. In addition, curcumin has antioxidant, anticarcinogenic, immunomodulatory, antifungal, and anti-inflammatory properties, which make it suitable in the medical and food industries [9,10,11,12,13,14,15]. However, the poor water solubility of curcumin limits its direct use. This poor solubility also leads to low absorption, fast metabolism, and quick systematic elimination [16,17]. To enhance the bioavailability and biological activity of curcumin, structural modifications are required [18].

Most research reported that stability can be improved by scaling to nanosize particles using high or low energy methods, such as ultrasound, high speed homogenization, or solvent evaporation [9,10,17,19]. Curcumin nanoparticles have a higher surface charge and surface area, are more hydrophobic, and have greater antioxidant activity than untreated curcumin. Curcumin nanoparticles have higher water solubility and suspension, which improves their antimicrobial activity [20,21,22].

The use of nanoemulsions is a promising technique for incorporating these bioactive complexes into foods [23,24,25]. Nanoscale curcumin provides superior properties in comparison to micron-scale curcumin, allowing it to mix effectively with other food ingredients to reduce the biological and enzymatic reactions, and allowing innovative products to be fabricated [26]. Nanoemulsions can be exploited in the food industry, since there are different approaches for their preparation, and they are stable systems for the encapsulation of bioactive substances [24,25,26]. Ultrasound techniques produce bioactive nanoemulsions with a higher yield and encapsulation efficiency [9,15,17,26,27].

The present study investigated the incorporation of curcumin in cheese in order to improve its dietary value. In this work, with the aid of ultrasound technology and polysorbate 20 as an emulsifier, a stable curcumin nanoemulsion (CUNE) was created. Data on particle size, encapsulation effectiveness, and stability were used to characterize the nanoemulsions. These were incorporated into the cheese which was checked, using a scanning electron microscope, against a control sample for sensory evaluation, antimicrobial activity.

## 2. Results and Discussion

Initially, we stabilized the CUNE containing edible oil and Tween 20 by using a sonochemical approach. To obtain stable CUNE, we optimized the oil and emulsifier concentration along with curcumin concentration. In the nanoemulsion, the curcumin concentration was encapsulated in the oil phase from 0.15 g to 0.75 g. We tried different inner and outer phase combinations to make a stable nanoemulsion, but only a few were found to be stable on centrifuge and heating at 80 °C for 30 min.

### 2.1. Characterization of Nanoemulsion

In the current study, we first used a sonochemical method to stabilize the curcumin nanoemulsion made up of edible oil and Tween 20. The curcumin concentration ranged from 0.75 g to 0.15 g in the oil phase of the current o/w nanoemulsion technology. Only a few of the inner and outer phase combinations we explored to create a stable nanoemulsion were discovered to be stable after centrifugation and 30 min of heating at 80 °C.

#### Particle Size Distribution, Polydispersity Index (PDI), and Optical Microscope for Stable Nanoemulsion

Nanotrac software was used to analyze the diameter size of the nanoemulsion and its polydispersity index (PDI) based on dynamic light scattering. Stable nanoemulsion results for the particle size and PDI are shown in Table 1, while stable particle size distribution is shown in Figure 1. With a PDI smaller than 0.4, stable nanoemulsions have a limited particle size distribution. According to Ahmed et al. (2012), an emulsion will not be stable if the PDI is higher than 0.5. In terms of their release phenomena, they also proposed that medium chain triglyceride-based nanoemulsions offer superior stability and improve curcumin’s bio-availability. According to Bagale et al., stable nanoemulsion with 50:50 (long-chain triglyceride: short-chain triglyceride) oil ratio has polydispersity less than 0.4 and an average droplet size of 200 nm. CUNE with medium-chain triglyceride palm oil has less water solubility than short-chain triglyceride oil, which shows a polydispersity index in the range of 0.3–0.357.

Figure 2a,b show the optical microscope image of CUNE with concentrations of 0.15 and 0.75 g curcumin. We can observe the encapsulation of curcumin in the oil in water morphology. The dark spots present in micelles are curcumin particles, which are surrounded by a layer of oil droplets in a stable form without any aggregation. There are some micelles without curcumin. The optical microscope images show the particle size distribution of the oil–water droplets. The droplet sizes are approximately 10–20 nm, calculated by using TEM and confirmed through light scattering. These oil droplets are tiny and disperse thoroughly in the nanoemulsion. The total phenolic content method used the normal correction curve of gallic acid at variance with an R^2^ value of 0.9894 to calculate encapsulation efficiency. CUNE has a higher encapsulation efficiency because of the small surface tension among droplets; flocculation, and accumulation are eluded, cultivating curcumin solubility. According to Bagale et al. (2022), who optimized the sonication period, polydispersity gives insight into the homogeneity of the size distribution and was low (0.3) for all samples, indicating the creation of monodisperse systems. This suggests that sonication would increase the effectiveness of encapsulation. Tween 20 concentrations that promote curcumin stability also increase the effectiveness of encapsulation [28].

Figure 3 shows the FTIR spectra of free curcumin and CUNE. The spectra of CUNE resembled that of pure curcumin, which contained all the normal absorption peaks. The two most significant functional groups were C=O stretching (1741.72 cm^−1^) and O-H bending (1348.24 cm^−1^). Due to the presence of O-C=O (2953.03 cm^−1^), O-H (2852.72 cm^−1^), and C-H (2922.16 cm^−1^) groups in the curcumin-loaded nanoemulsion, the other bands in the area of 2965–2855 cm^−1^ were provided by the C-H stretching vibration.

### 2.2. Transmission Electron Microscope Analysis of Nanoemulsion

Figure 4 shows that the nanoemulsion has a particle size of 14 nm and is spherical in shape. According to Biswas et al. [20], the noble assets and functionality of the nanoparticles are owing to his higher ratio surface area/volume and nanoscale size and hydrophobicity. They also reported that CUNE with a particle size of approximately 5–50 nm shows higher antimicrobial potential than its natural form.

### 2.3. Antimicrobial Activity Nanoemulsion

The outcomes of the antibacterial potential of CUNE are revealed in Table 2 (Figure 5). CUNE with a concentration of 100 and 50 μg/mL showed the highest antibacterial activity against S. aureus and E. coli. At a concentration of 25 μg/mL, the nanoemulsion showed reasonable antibacterial activity for both strains; however, the lowest antimicrobial inhibition level was at a diameter of 5 mm. An earlier study found that CUNE had better aqueous-phase solubility and dispersibility than pure curcumin and hence had antibacterial activity. Any nanoscale particle’s antibacterial potential will depend on its physicochemical characteristics (size, shape, and surface qualities), as well as the quantity used. According to Naghadri et al. [29] and Wang et al. [30], nanoparticles smaller than 100 nm have a higher adhesion to the surface of cell membranes than larger nanoparticles, which can cause disruptions in the functions of the cell membrane.

### 2.4. Curcumin Nanoemulsion in Cheese Formulation and Its Analysis

Based on the above result, we took the curcumin nanoemulsion (CUNE 2) for further study in terms of its addition in cheese formulation.

#### 2.4.1. Sensory Analysis of Cheese

The number of foods enriched with bioactive compounds are increasing in the dairy and food industries. As bioactive compounds show better antioxidant activity, these food products promote health and wellness. Figure 6 provides an evaluation of the CUNE-enriched effect on the organoleptic properties of cheese compared to the control and emulsion cheeses during a 60-day period (at 4 °C).

With a decrease in the particle size, there is an increase in the surface-to-volume ratio, and a reduction of the concentration at which compounds are added to food products. This does not hamper its sensory analysis compared to the control sample. The data show that lower concentrations can have more strength than otherwise [31], such as a droplet delivering the identical sensory profile [32] or an increase in sensory perception [33]. It is suggested that a larger particle surface causes an increase in the saliva dissolution rate and, possibly, a stronger taste perception. Figure 6 shows that there were no significant differences (*p* > 0.05) in color, smell, appearance, acceptability, or taste between the control and CUNE-enriched samples. However, when compared to the control and CUNE samples, the emulsion without curcumin samples had the lowest (*p* < 0.05) sensory values. This is owing to the structural change in the cheese matrix caused by the larger particle size of the emulsion affecting the organoleptic values. While the outer layer of a nanoemulsion can be designed to be resistant to the environment during the first stages of ingestion, preventing consumers from experiencing any unpleasant tastes or odors, nanoencapsulation may offer a potential mechanism to physically trap the compounds that cause these unpleasant tastes and odors. The product’s quality and acceptability are directly tied to the raw material and production quality. Consumers are showing a preference for minimally processed foods with the highest natural constituents, additives with health benefits such as antimicrobials, and naturally occurring antioxidants. In order to meet this demand, nanotechnology allows for the development of many ingredients’ functionality by lowering the concentration of substances, altering their solubility, and enhancing or controlling their effectiveness [31,32].

#### 2.4.2. Physicochemical Parameters for Cheese

The physicochemical features of nutriments play a part in defining their effectiveness and shelf life. Table 3 shows that the physicochemical properties of the cheese treated with two types of emulsifiers (CUNE and emulsion) compared to the control sample stored at 4 °C and kept for 30 and 60 days. With the fresh sample as a control, the CUNE- and emulsion-based cheeses do not have significant differences (*p* > 0.05) for dry matter, ash, and protein content. However, after 30 days, there are significant difference (*p* < 0.05) observed in the percentage of dry matter for CUNE and emulsion compared to the control, which increased in dry matter content. As dry matter content increases, the total phenolic content decreases and shows less antioxidant activity. Whereas the CUNE sample has less dry matter content over time, which maintains its integrity in the cheese matrix, as the nanoemulsion, with a smaller particle size, is better distributed within the cheese matrix. Fresh samples from each group did not differ significantly from one another in the content of ash, fat, or protein (*p* > 0.05), while the control samples showed significant differences (*p* < 0.05) after 30 and 60 days of storage. The percentage of fat, protein, and ash increased in the control sample, while the CUNE and emulsion samples showed no significant differences [34,35]. The presence of emulsifiers in the treated samples reduced the increase of dry matter and thus reduced moisture loss compared to the control sample. Thus, the nanoemulsion is recommended for fortifying cheese. For the pH, there was no noticeable change in the pH values with a storage period of 30 or 60 days.

### 2.5. Scanning Electron Microscope Analysis of Cheese

SEM was used to study the structural morphology of the cheese matrix with the addition of CUNE and the emulsion in comparison to the control. Three-dimensional images were obtained to facilitate the identification of substances. The photograph of the sample (control samples 1, 2) is shown in Figure 7. There are fewer variances in the protein structure between the CUNE samples compared to the control. There is some free fat present, and the cheese fat and protein clusters are tightly packed. The control sample’s protein and fat clusters look smaller than those in the CUNE sample. The control sample has substantially more free fat globules than those seen in the other samples. The fat-free globules in the control sample are smaller than those in the emulsion sample. The hardness was reduced, and the casein network structure softened when CUNE was added, showing that the incorporation of CUNE transformed the structure of the protein–fat network. CUNE was added together with emulsifying salts, which increased the dispersion of fat globules. It is possible to infer that the cheese produced with the CUNE supplement has particle-filled gel networks, where fat globules serve as filler molecules in the protein network. According to [31], foods containing nanoemulsions enhance the functionality of the formulations’ constituents. Nanoemulsions can also be utilized to modify texture. Depending on the internal-phase percentage, oil composition, stabilizer (type and concentration), and droplet size, nanoemulsions may exhibit rheological characteristics that are different from those of viscous liquids in viscoelastic solids.

### 2.6. Antioxidant Activity of Curcumin Nanoemulsion

As shown in Table 4, the antioxidant capacity of the control sample and samples 1 and 2 is expressed as µg of Trolox equivalents (TE) per g of nanoemulsion. Regardless of surfactant content, the antioxidant capacity of the nanoemulsion as determined by the ferric reducing antioxidant power (FRAP) assay did not show any significant changes. The antioxidant capacity values reported by the FRAP assay, however, were considerably lower than those obtained by DPPH. In this respect, it follows that if one species is reduced, another must be oxidized. Curcumin’s three active sites can be oxidized through hydrogen abstraction and electron transfer. Test samples differed in antioxidant activity depending on the additives introduced into the formulation. Samples 1 and 2, compared with the control sample, had a higher antioxidant activity. The CUNE cheese (sample 2) showed the most significant antioxidant activity (2.3 times the control sample). CUNE contributes to maintaining and enhancing BAS (curcumin) encapsulated in the nanoemulsion with sonication. This is due to the fact that the antioxidant capabilities of the nanoemulsions containing Tween 20 differed depending on the surfactant concentration and the assay for curcumin measurement utilized. Although based on our findings that high Tween 20 concentrations resulted in a slow release of curcumin, it has been reported that an excess of this surfactant can form micelles capable of encasing the bioactive compound, which enhances the protection of encapsulated curcumin and thereby boosts the system’s antioxidant capacity. The sonication method, which compactly embeds curcumin in the nanoemulsion and controls its solubility, is one of the main reasons for the increase in antioxidant activity.

## 3. Materials and Methods

Native lab grade curcumin (Magiya vostoka), Safflower oil (FAO Code: 0281), Tween 20 (DIY ECO Cosmetic, 3.2 fat% milk (Prostokvashino) and NaCl were procured from a local market in Chelyabinsk, Russian Federation. In all experiments, a 440 W U-sonic ultrasound at 80% power, 22 ± 1.65 kHz, and with a tip diameter of 22 mm. Distilled water was used for the experiment.

### 3.1. Ultrasound Assisted Curcumin Encapsualtion

Initially, to prepare a lipid phase of the emulsion, curcumin powder (0.05 g/mL) was dissolved in Safflower oil at 50 °C [15]. The aqueous phase was created by dissolving Tween 20 in distilled water at concentrations of 0.2 and 0.3 g/mL oil. The oil phase was added slowly to the water phase and stirred to make a coarse emulsion (15 min at 2000 rpm), then sonochemically at 80% amplitude of the total power (400 W). A digital thermometer was used to track the temperature during the emulsification and to keep it below 50 °C. A cool water jacket encircled the reactor, and the sonication time was divided into four cycles of 3 min each to reduce hot spot creation during sonication. Once the sample was prepared under vacuum, the emulsion was freeze dried (Figure 8).

### 3.2. Physicochemical Analysis of Nanoemulsions

#### Nanoemulsion Stability

Nanoemulsion stability was confirm using a technique defined in Kumar et al. 2016 [28]. The 10 mL of nanoemulsions samples were kept in a hot water bath at 80 °C for 30 min, then moved to a freeze for 15 min, and monitored by centrifugation (Hettich Zentrifugen, Mikro 22R) at 5000 rpm for 30 min. The whole volume (WV) and emulsion phase volume (EPV) of the nanoemulsion in the centrifuge tube were measured. The nanoemulsion stability was calculated using the formula below (ES):(1)$$\mathrm{Nanomulsion}\;\mathrm{stability}\;\left(\%\right)=\frac{\mathrm{Volume}\;\mathrm{of}\;\mathrm{nanoemulsion}\;\mathrm{phase}}{\mathrm{Total}\;\mathrm{volume}\;\mathrm{of}\;\mathrm{nanoemulsion}}\times 100$$

Using Nanotrac FLEX, the average particle size of the fresh nanoemulsion was evaluated. For analysis, the nanoemulsion sample was diluted in water to measure the particle size through dynamic light scattering. The total phenolic content of the nanoemulsion was used to calculate encapsulation efficiency. The Folin–Ciocalteu reagent was used to determine the total phenolic content of the nanoemulsion. The percentage oxidation inhibition and encapsulation effectiveness of the CUNE were measured before and 30 min after the centrifugation at 5000 rpm. Encapsulation efficiency was determined using the method in [36] with some modifications. For instance, 15 mL of the nanoemulsion was centrifuged at 5000 rpm at 5 °C for 30 min after being passed through a filter membrane. Following the collection of centrifuged permeate, the UV (Shimadzu UV-2700, Japan) absorbance at 520 nm wavelength was measured. Triplets were performed for all the measurements. For optical imaging of the nanoemulsion, a sample on a slide was dried and checked at magnifications of 40× to 60×. CUNE and free curcumin scanned in the range of 400–4000 cm^−1^ wavelength using a Fourier-transform infrared (FTIR) spectrometer.

### 3.3. Transmission Electron Microscope (TEM) of Nanoemulsion Samples

The magnitude and nature of the CUNE were resolute by TEM (JEOL-100 CX). An aqueous solution of nanoemulsion sample (1:4 ratio) was sonicated for 10 min. A droplet of the sample was placed on a 200-mesh carbon-coated copper grid at room temperature and dried, before 2% uranyl acetate was added at 37 °C and grid mounted for TEM inspection.

### 3.4. Antimicrobial Activity of CuNE

For the antimicrobial potential of CUNE, we used two different strains: one positive strain, *Staphylococcus aureus*, and one negative strain, *Escherichia coli*. To check antimicrobial activity, we performed a zone inhibition test. The bacteria culture was added to a Petri dish that contained nutrient agar. On the nutrient agar, was a sample that had been antimicrobially treated. The Petri dish was then kept at 36 °C for 18 to 24 h.

### 3.5. Preparation of Cheese with Nanoemulsion

The ultrafiltrate (UF) milk retentate was pasteurized (72 °C, 15 s) and then a 5% nanoemulsion with and without curcumin were enriched for cheese production at 32 °C. A standard white cheese was produced and used as a reference. The schematic formation of the cheese is shown in Figure 9.

#### Physical and Chemical Analysis of Nanoemulsion-Enriched Cheese

The pH of the nanoemulsion-based cheese was restrained through an electrode (model HI98103, Hanna Instruments, Romania) inserted into grated cheese after calibration with the standard buffers pH 4 and 7 from 22 to 31 °C. Titratable acidity was measured (g/100 g of lactic acid) using the technique in [37]. Dry matter (DM) content of cheese samples was analyzed using the oven drying method at 102 ± 2 °C. The Kjeldahl method determined total nitrogen (TN), WSN, and NPN (Jalilzadeh et al. 2020). The cheese was cut into cylinders at a height of 20 mm and a diameter of 20 mm, using a stainless-steel cylinder knife, and kept at room temperature (20 °C). The ash content of the cheese samples was determined using the method described in [38].

### 3.6. Field Emission Scanning Electron Microscope of Cheese

Cheese samples of 5 mm^2^ had their exterior microstructure and morphology determined by FE-SEM. The cheese was immersed into liquid nitrogen to remove any moisture content and then ruptured. Using an ion sputter, gold was sucked into the punctured cheese. Finally, the sample was imaged using FE-SEM at 5 kV (Jeol JSM-7001F, Moscow).

### 3.7. Organoleptic Analysis of Cheese

We invited 15 students (aged 25–40) from the food and biotechnology department of South Ural State University, Chelyabinsk for the organoleptic analysis of the cheeses. Each evaluator received the three kinds of cheese simultaneously, each with a different number assigned to it. The panel consisted of 15 experts who rated the smell, color, appearance, mouth feel, and taste in two consecutive sessions using shapeless scales with anchors at the split ends. A nine-point system was used (1 = extremely dislike, 3 = moderately dislike, 5 = neither like nor dislike, 7 = moderately like, and 9 = extremely like).

### 3.8. Antioxidant of Processed Cheese

A 0.1 mm DPPH radical solution was made to test the antioxidant activity of the cheese samples with CUNE. Using a UV spectrophotometer, the solution’s absorbance was measured at 515 nm. Cheese samples were prepared by soaking 10 g in 90% ethanol for 45 min at 150 rpm in a LOIP LS-120 laboratory shaker. The mixture was then centrifuged for 10 min, and the supernatant was collected for testing. Each sample received 280 µL of DPPH radical solution and 20 µL of supernatant in a microplate. After 30 min of incubation, the samples were examined, and the absorbance was calculated using a 517 nm reference wavelength.

### 3.9. Statistical Analysis

For each group, the data are shown as the standard error of mean (SEM). GraphPad Prism software version 8.0 was used for statistical analysis (GraphPad software, 2019). The column values in Table 1, Table 2, Table 3 and Table 4 were with significance of difference (*p* < 0.05).

## 4. Conclusions

The current work demonstrates the successful formation of stable curcumin encapsulation in a nanoemulsion with a high percent of loading (0.05 g/mL) containing Tween 20 surfactant using the ultrasound approach and its incorporation in cheese. CUNE shows good antimicrobial activity for *S. aureus* and *E. coli* (12- and 18-mm zone inhibition diameters). Based on the data, we can conclude that there is no negative impact of CUNE on cheese, as it maintain its overall sensory analysis and physicochemical properties in comparison with a control sample. A normal emulsion without curcumin shows reasonable sensory analysis compared to control emulsion.

## Figures and Tables

**Figure 1 ijms-24-02663-f001:**
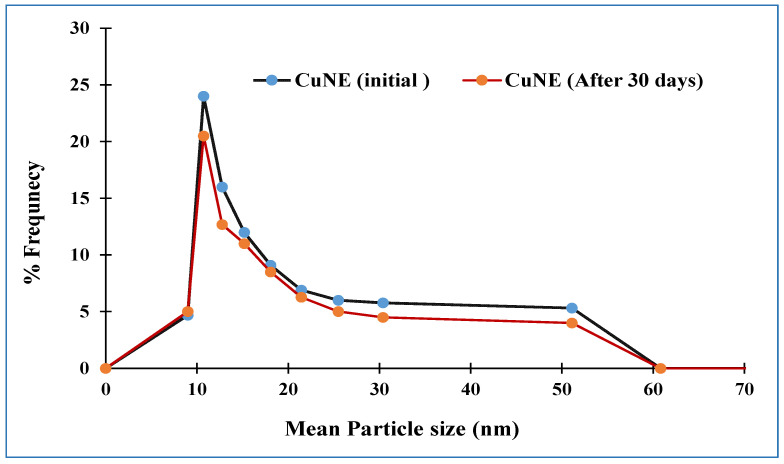
Stability data of curcumin nanoemulsion sample in term of PSD.

**Figure 2 ijms-24-02663-f002:**
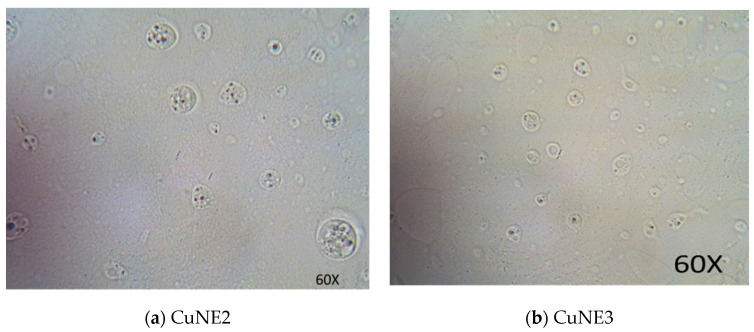
Optical image for curcumin encapsulation in different nanoemulsion sample.

**Figure 3 ijms-24-02663-f003:**
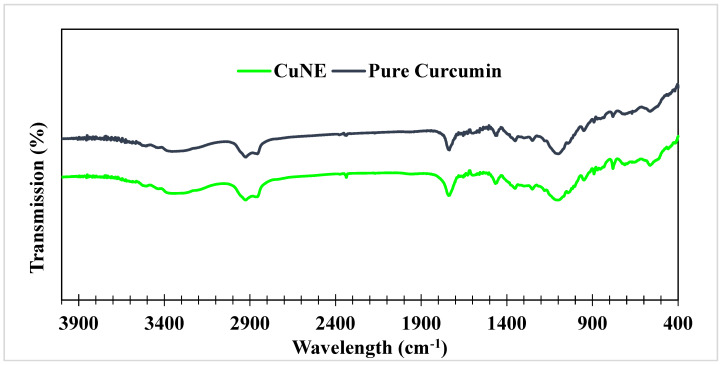
FTIR spectra Free curcumin and Curcumin nanoemulsion.

**Figure 4 ijms-24-02663-f004:**
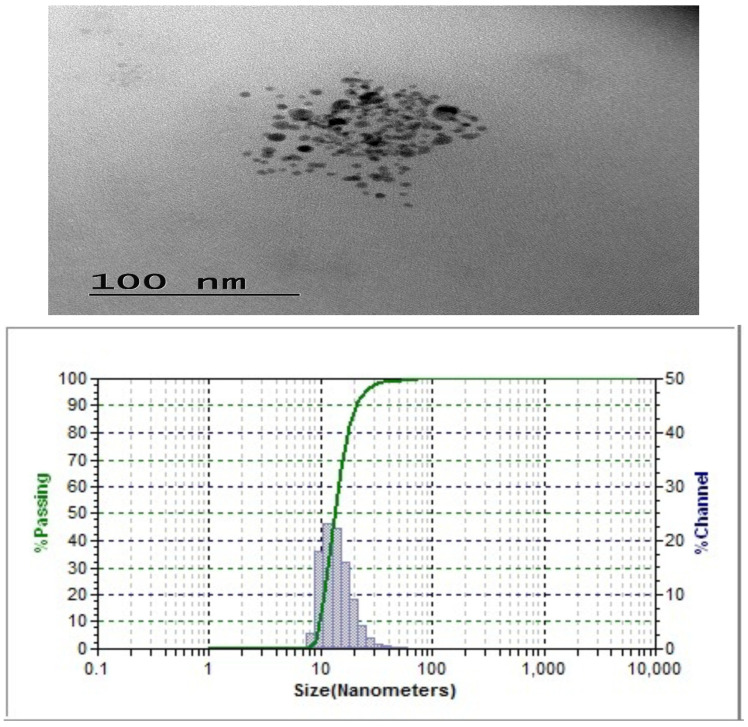
Transmission electron microscope analysis of CUNE sample.

**Figure 5 ijms-24-02663-f005:**
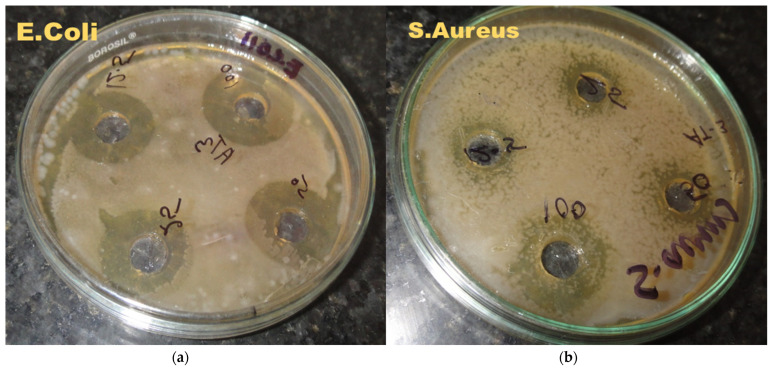
Antimicrobial activity of curcumin nanoemulsion for (**a**) *E. coli* and (**b**) *S. aureus*.

**Figure 6 ijms-24-02663-f006:**
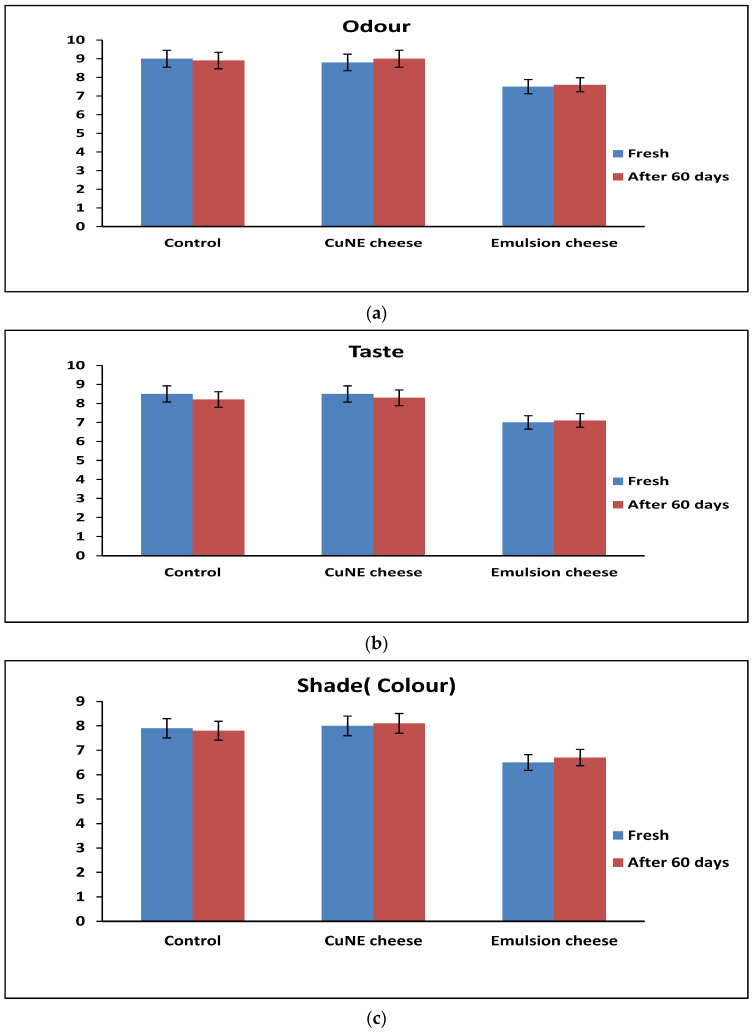
Sensory analysis of cheese sample with addition of curcumin nanoemulsion and emulsion: (**a**) Odor, (**b**) Taste, (**c**) Shade, (**d**) Appearance, and (**e**) Acceptability.

**Figure 7 ijms-24-02663-f007:**
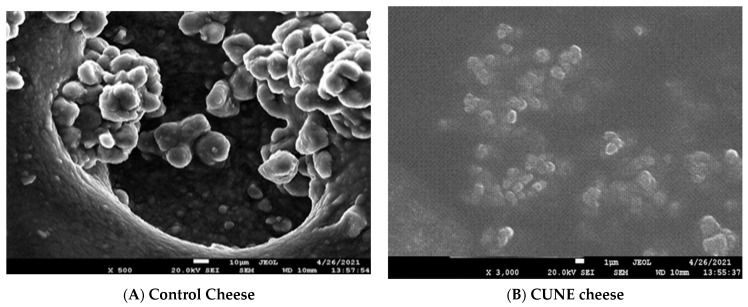
Photo study experienced samples prepared using scanning electron microscopy (total magnification of ×5000) Cheese matrix: (**A**) Control, (**B**) CUNE (**C**) Normal emulsion.

**Figure 8 ijms-24-02663-f008:**
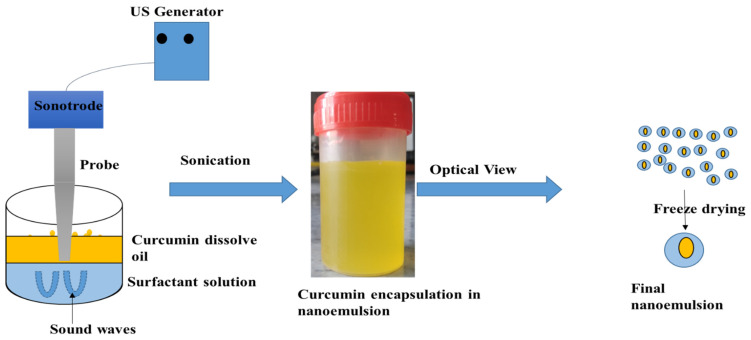
Schematic diagram for preparation of curcumin nanoemulsion using ultrasound.

**Figure 9 ijms-24-02663-f009:**
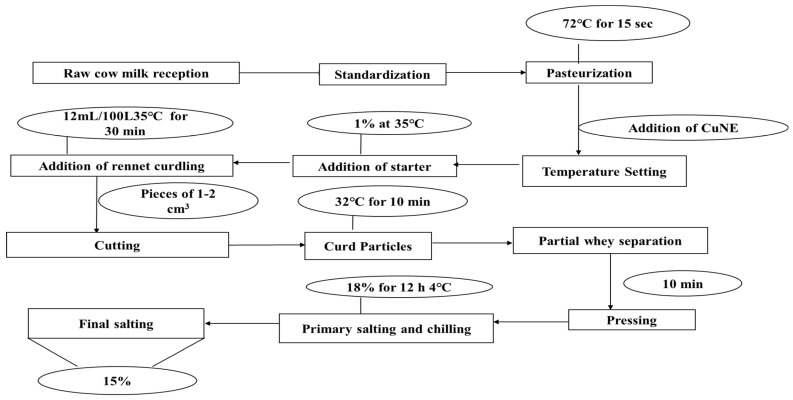
Schematic diagram for addition of curcumin nanoemulsion during cheese making.

**Table 1 ijms-24-02663-t001:** pH and gravitational stability of nanoemulsions data.

Exp No	pH	Gravitational Stability at Room Temperature(Days)	Particle Size for Fresh Sample (nm)	Particle Size after 60 Days (nm)	PDI	Zeta Potential mV	Encapsulation Efficiency
CUNE 1	5.68	15	180 ± 0.5	820 ± 5	0.63	−15	94%
CUNE 2	5.62	70	13.35 ± 0.5	14.05 ± 0.5	0.27	−25	
CUNE 3	5.69	75	11.05 ± 0.5	12 ± 05	0.35	−27	
CUNE 4	5.55	15	135 ± 3	625 ±10	0.43	−18	

**Table 2 ijms-24-02663-t002:** Antimicrobial activity of CUNE.

Microorganism	Zone Inhibition Diameter (mm)			
	100 *	50 *	25 *	12.5 *
E. Coli	18	15	13	8.2
S. Aureus	12	9	6	3

Concentration of CUNE are in percent as follows 100 * = 100 µg NC/mL, 50 * = 50 µg NC/mL, 25 * = 25 µg NC/mL, and 12.5 * = 12.5 µg NC/mL. Note: All analysis measurement was conducted in triplets and have ± SEM (*p* < 0.05).

**Table 3 ijms-24-02663-t003:** Physicochemical analysis of cheese.

	Storage Time	Ash,%	Dry Matter,%	pH	Protein,%	Fat,%
Control	Fresh	5.53 ± 0.13	39.46 ± 0.34	6.2 ± 0.08	12.12 ± 0.08	18.85 ± 0.27
	30 days	5.98 ± 0.14	42.34 ± 0.11	6.04 ± 0.04	12.66 ± 0.09	19.65 ± 0.2
	60 days	6.01 ± 0.1	43.24 ± 0.15	6.01 ± 0.06	12.91 ± 0.12	20.71 ± 0.13
E1 (Cur)	Fresh	5.66 ± 0.11	39.28 ± 0.33	6.19 ± 0.08	12.63 ± 0.36	18.68 ± 0.23
	30 days	5.71 ± 0.14	40.53 ± 0.11	6.05 ± 0.04	12.6 ± 0.09	18.93 ± 0.38
	60 days	5.75 ± 0.08	41.72 ± 0.22	6.03 ± 0.03	12.68 ± 0.09	19.14 ± 0.12
E2	Fresh	5.61 ± 0.38	39.14 ± 0.57	6.21 ± 0.08	11.72 ± 0.15	18.97 ± 0.27
	30 days	5.64 ± 0.07	41.41 ± 0.48	6.1 ± 0.01	11.7 ± 0.17	19.23 ± 0.21
	60 days	5.69 ± 0.09	41.84 ± 0.22	6.04 ± 0.05	11.74 ± 0.14	19.41 ± 0.12

**Table 4 ijms-24-02663-t004:** AOA prototypes cheese.

Specimen	AOA Prototypes Crumb mg
The control	0.1059 ± 0.05
Sample 1 (CUNE)	0.3164 ± 0.015
Sample 2 (Emulsion)	0.0829 ± 0.02

## Data Availability

The datasets used and/or analyzed during the current study are available from the corresponding author on reasonable request.

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
