# Peer review of "Prospect of Bioactive Curcumin Nanoemulsion as Effective Agency to Improve Milk Based Soft Cheese by Using Ultrasound Encapsulation Approach"

_ijms, 2023, doi:10.3390/ijms24032663_

Round 1
Reviewer 1 Report
The authors reported curcumin nanoemulsion to improve milk-based soft cheese. The reported issues have relevance in nanoliposome technology for the food industry. This manuscript can be considered for publication in this Journal after major revisions.
1- Enhance the image quality of Figure 2.
2- Please consider the results of FTIR analysis of curcumin nanoemulsion (Cu-NE) in section 2.2 Physicochemical analysis of nanoemulsions.
3- Pictures of inhibition zones of the standard disc against S. Aureus and E. coli. should have been presented in section 4.3 Antimicrobial activity nanoemulsion.
4- Since you have examined the antibacterial properties, it is better to present the results of the In vitro release of curcumin from the nanoparticles.
5- The TEM image (Figure 5) does not show interesting results. Please repeat this test.
Author Response
Response to Reviewer 1
Thank you very much to Reviewer for their valuable comment on manuscript, it will be helpful to upgrade the manuscript according journal requirement
Comments and Suggestions for Authors
- Enhance the image quality of Figure 2.
Answer: We are very thankful to the reviewer for this comment, it will enhance the figure quality of revised version of manuscript. We incorporate the required change in figure 2 as per comment
2- Please consider the results of FTIR analysis of curcumin nanoemulsion (Cu-NE) in section 2.2 physicochemical analysis of nanoemulsions.
Answer: We are very thankful to the reviewer for this comment, it will enhance the quality of revised version of manuscript. We incorporate the required change in section 2.2 as well as result discussion section 4.1(Figure 5 as per revised manuscript) as per comment.
Figure 5 shows the FTIR spectra of free curcumin and CUNE. The spectra of CUNE resembled that of pure curcumin, which contained all the normal absorption peaks. The two most significant functional groups are C=O stretching (1741.72 cm-1) and O-H bending (1348.24 cm-1). Due to the presence of O-C=O (2953.03 cm-1), O-H (2852.72 cm-1), and C-H (2922.16 cm-1) groups in the curcumin-loaded nanoemulsion, the other bands in the area of 2965–2855 cm-1 were provided by the C-H stretching vibration.
3- Pictures of inhibition zones of the standard disc against S. Aureus and E. coli. should have been presented in section 4.3 Antimicrobial activity nanoemulsion.
Answer: We are very thankful to the reviewer for this comment, it will enhance the quality of revised version of manuscript. We incorporate the required change in section 4.3 (Figure 7 as per revised manuscript)
4- Since you have examined the antibacterial properties, it is better to present the results of the In vitro release of curcumin from the nanoparticles.
Answer: We are very thankful to the reviewer for this comment, it will enhance the quality of revised version of manuscript. However in our previous study U. Bagale, A. Tsaturov, I. Potoroko, S. Potdar, S. Sonawane, In-vitro evaluation of high dosage of curcumin encapsulation in palm-oil-in-water, nanoemulsion stabilized with a sonochemical approach, Karbala Int. J. Mod. Sci. 8 (2022) 83–95. https://doi.org/10.33640/2405-609X.3205, we presented the result so we thought to not repeat the analysis again for current manuscript.
5- The TEM image (Figure 5) does not show interesting results. Please repeat this test.
Answer: We are very thankful to the reviewer for this comment, it will enhance the quality of revised version of manuscript. We incorporate the required change in section 4.2 (Figure 6 as per revised manuscript).

Reviewer 2 Report
The article reported the nanoemulsion of curcumin with ultrasound encapsulation approach to improve the solubility in diary products. The article needs to be improved according comments below
Line 58, a grammatical error was found. Authors need to check
Line 59 and 62, The use of word “we” should be avoided.
Line 65-67 should have sub-heading “Materials”. Tween 20 and Milk should be written completely with the brand. It is important to know the quality of the products and what type of milk was used. Grammatical error was found, the past tense should be used, not present tense.
There are a few unclear phrases and sentences in Line 70-77. For example “….. and stirrer for make an coarser…..”., The verb of the sentences is not clear. It happened also in the last sentence in the paragraph.
Line 83. “… defined in [28]. The citation should state the name and the year “… defined in name, year..”
Figure 1. Quality of the figure needs to be improved. In Final nanoemulsion, there is a blue circle. what is that?
The arrow is not clear. It seems after the O/W emulsion, the process is continued to encapsulation. The clarity of the figure needs to be improved.
In subsection Nanoemulsion stability, author wrote unclear measurements and sentences. How the total volume and phase volume was measured. What does “it” referr to in Line 92. Line 93, total phenolic content needs to be measured, but the previous sentences talk the average particle sizes, which is not relevant with total phenolic content.
Line 103, “resolute” should be “resoluted”
last sentences in Section 3 is not formed as a sentence.
Line 109-110, the name of microbes should be in “italic”.
In methods, there are a lot of grammatical error. All should be in past tense, not present tense.
Line 116, UF (ultafiltrate). The extended word first and then followed by the abbreviation
What is (72, 15 sec) in line 116?
In general, there are a lot of common mistakes (grammars, low quality figure and presentation, unclear sentences”.
The article needs to be improved in the whole manuscript
According the comments, the article is suggested to be “rejected”
Author Response
Response to Reviewer 2
Thank you very much to Reviewer for their valuable comment on manuscript, it will be helpful to upgrade the manuscript according journal requirement
- Line 58, a grammatical error was found. Authors need to check
Answer: We are very thankful to the reviewer for this comment, it will enhance the quality of revised version of manuscript. We incorporate the required change as per comment
- Line 59 and 62, The use of word “we” should be avoided.
Answer: We are very thankful to the reviewer for this comment, it will enhance the quality of revised version of manuscript. We incorporate the required change as per comment
- Line 65-67 should have sub-heading “Materials”. Tween 20 and Milk should be written completely with the brand. It is important to know the quality of the products and what type of milk was used. Grammatical error was found, the past tense should be used, not present tense.
Answer: We are very thankful to the reviewer for this comment, it will enhance the quality of revised version of manuscript. We incorporate the required change as per comment, we made changes on whole material and method sections.
- There are a few unclear phrases and sentences in Line 70-77. For example “….. and stirrer for make an coarser…..”., The verb of the sentences is not clear. It happened also in the last sentence in the paragraph.
Answer: We are very thankful to the reviewer for this comment, it will enhance the quality of revised version of manuscript. We incorporate the required change done on whole section of material and method.
- Line 83. “… defined in [28]. The citation should state the name and the year “… defined in name, year..”
Answer: We are very thankful to the reviewer for this comment, it will enhance the quality of revised version of manuscript. We incorporate the required change as per comment
- Figure 1. Quality of the figure needs to be improved. In Final nanoemulsion, there is a blue circle. what is that?
The arrow is not clear. It seems after the O/W emulsion, the process is continued to encapsulation. The clarity of the figure needs to be improved.
Answer: We are very thankful to the reviewer for this comment, it will enhance the quality of revised version of manuscript. We incorporate the required change as per comment. Figure 1 now revised with better quality along with proper schematic.
- In subsection Nanoemulsion stability, author wrote unclear measurements and sentences. How the total volume and phase volume was measured. What does “it” referr to in Line 92. Line 93, total phenolic content needs to be measured, but the previous sentences talk the average particle sizes, which is not relevant with total phenolic content.
Answer: We are very thankful to the reviewer for this comment, it will enhance the quality of revised version of manuscript. We incorporate the required change as per comment. Yes we agree that particle size measurement is not relevant to the total phenolic content, however we just provide possible measurement method one after another.
- Line 103, “resolute” should be “resoluted”
Answer: We are very thankful to the reviewer for this comment, it will enhance the quality of revised version of manuscript. We incorporate the required change as per comment.
- last sentences in Section 3 is not formed as a sentence.
Answer: We are very thankful to the reviewer for this comment, it will enhance the quality of revised version of manuscript. We incorporate the required change as per comment.
- Line 109-110, the name of microbes should be in “italic”.
Answer: We are very thankful to the reviewer for this comment, we incorporate the required change as per comment.
- In methods, there are a lot of grammatical error. All should be in past tense, not present tense.
Answer: We are very thankful to the reviewer for this comment, it will enhance the quality of revised version of manuscript. We incorporate the required change done on whole section of material and method.
- Line 116, UF (ultafiltrate). The extended word first and then followed by the abbreviation
Answer: We are very thankful to the reviewer for this comment, we incorporate the required change in revised manuscript.
- What is (72, 15 sec) in line 116?
Answer: We are very thankful to the reviewer for this comment, we incorporate the required change in revised manuscript. (72 °C, 15 sec)
- In general, there are a lot of common mistakes (grammars, low quality figure and presentation, unclear sentences”.
Answer: We are very thankful to the reviewer for this comment, it will enhance the quality of revised version of manuscript. We incorporate the required change on whole manuscript with extensive English correction and with proper statement.

Round 2
Reviewer 1 Report
The experiments, results and discussion as well as supporting information are well presented and this paper is ready for publication in every respect.
Author Response
Thank you very much for reviwer for their timely comment on manuscript.
Reviewer 2 Report
Thank you for comment and improvements. However, Figures are still required for improvement.
Figure 1, 2, 6 has blur font.
Figure 3 is not balance, too wide.
Table 3 needs to be improved.
Quality of figure needs to be checked. It should use high quality figure.
Author Response
Thank you very much for reviwer for their valuble suggestion for this manucript
